# Precise Robotic Needle-Threading with Tactile Perception and Reinforcement Learning

**Zhenjun Yu\*, Wenqiang Xu\*, Siqiong Yao, Jieji Ren,**
**Tutian Tang, Yutong Li, Guoying Gu, Cewu Lu§**
Shanghai Jiao Tong University

**Abstract:** This work presents a novel tactile perception-based method, named T-NT, for performing the needle-threading task, an application of deformable linear object (DLO) manipulation. This task is divided into two main stages: *Tail-end Finding* and *Tail-end Insertion*. In the first stage, the agent traces the contour of the thread twice using vision-based tactile sensors mounted on the gripper fingers. The two-run tracing is to locate the tail-end of the thread. In the second stage, it employs a tactile-guided reinforcement learning (RL) model to drive the robot to insert the thread into the target needle eyelet. The RL model is trained in a Unity-based simulated environment. The simulation environment supports tactile rendering which can produce realistic tactile images and thread modeling. During insertion, the position of the poke point and the center of the eyelet are obtained through a pre-trained segmentation model, Grounded-SAM, which predicts the masks for both the needle eye and thread imprints. These positions are then fed into the reinforcement learning model, aiding in a smoother transition to real-world applications. Extensive experiments on real robots are conducted to demonstrate the efficacy of our method. More experiments and videos can be found in the supplementary materials and on the website: https://sites.google.com/view/tac-needlethreading.

**Keywords:** tactile perception, needle threading

## 1  Introduction

Deformable linear object (DLO) insertion is a common task in everyday life. From needle threading to suturing in medical surgery scenarios. In this work, we are particularly interested in the needle-threading task as it requires precisely manipulating the highly deformable thread and locating the target eyelet with a small clearance, which is challenging, even for humans.

The task of needle-threading can be subdivided into two stages: *Tail-end Finding* and *Tail-end Insertion*. During the *Tail-end Finding* stage, the primary objective of the agent is to locate and secure the terminal end of the thread. Compared to the thread, it is ostensibly less complex to identify the tail-end of other DLOs such as network and USB cables. The tail-ends of these DLOs are easily distinguished from the rest of their line part and are often sufficiently large to be detected by a standard RGB-D sensor from its working distance. Conversely, the tail-end of a thread bears a uniform appearance with the remainder of the thread, rendering it virtually invisible to a standard camera lens. In the *Tail-end Insertion* stage, the agent should accurately guide the tail-end of the thread into a specified eyelet with minimal clearance. In addition, the agent should be able to judge the success or failure of the task execution.

Based on the observations, we propose a **T**actile perception-based approach to address the **N**eedle-**T**hreading task and name it **T-NT**. In comparison with previous solutions which adopt laser scanner [1], high-resolution camera [2], and dual cameras [3] to locate and discern the states of both the

---

\* indicates equal contributions.      § Cewu Lu is the corresponding author.

7th Conference on Robot Learning (CoRL 2023), Atlanta, USA.

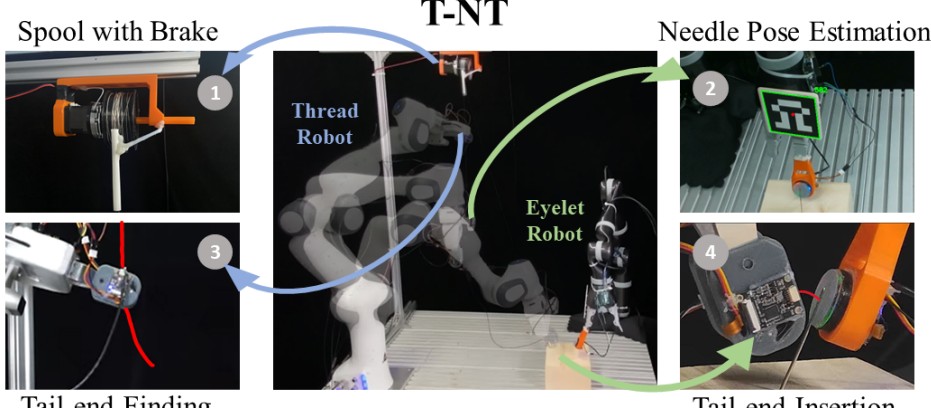

Figure 1: Our task setting contains one thread robot with a RealSense camera for marker detection and one eyelet robot. There are two stages for our task, *Tail-end Finding* and *Tail-end Insertion*. The brake is connected to the scroll to control whether it can rotate or not.

thread and eyelet, our approach utilizes the tactile sensor which is locally mounted on the gripper fingers. We make use of MC-Tac [4], a GelSlim[5]-like tactile sensor, to provide tactile perception. The tactile imprints produced by the thread and eyelet are distinctly observable when they are pressed on the gel of the tactile sensor.

To find the tail end, the agent is instructed to trace the contour of the thread twice. In the first run, the goal is to measure the distance from the starting point to the extremity of the thread. The endpoint of the thread imprint within the tactile image serves as a determinant of the gripper reaching the tip. It is crucial to note that, due to the pliable nature of the thread, it can get straightened as the gripper glides over it. Thus, once the robot gripper grasps the starting point, it only needs to follow a linear path. The total thread length can be approximated by cumulatively adding the distance traversed by the gripper and the remaining length of the thread imprint visible on the tactile image. For the second run, the agent initiates from the identical starting point, proceeding to the terminal end. This time the gripper stops at a certain distance from the tip. This approach ensures that the tail-end is grasped with a minimal residual segment ready for insertion into the eyelet.

The *Tail-end Insertion* process is facilitated by employing a tactile-guided reinforcement learning (RL) model. The task is conceptualized as a goal-conditioned RL task, wherein the objective is for the thread's tip to poke the gel at the back of the eyelet and the poke point is within the eyelet's range. To accomplish this, an RL model is trained from a simulated environment built on the Unity platform and deployed to the real world. Tactile rendering is implemented in the simulator. It can produce tactile images that resemble the images from the real-world sensor. The thread is represented as chaining particles constrained by distance and bend effect, modeled by XPBD (Extended Position-based Dynamics)[6]. To obtain the position of the poke point and the eyelet, we adopt a pretrained language-grounded segmentation model, Grounded-SAM [7, 8] to predict the masks for both the needle eye and thread imprints. And then we calculate the center of mass for the poke point and sample several pixels to represent the eyelet area based on the masks. We feed the position information of the poke point and the eyelet as inputs to train the reinforcement learning model, ensuring a smoother transition to real-world applications.

We conduct experiments with needles of different sizes and threads of different sizes. We have achieved an average 63.33% success rate in the real world.

We summarize our contribution as follows:

- We propose a tactile perception-based approach for the needle threading task, T-NT, with two stages, *Tail-end Finding* and *Tail-end Insertion*.
- We conducted real experiments on different kinds of threads and needles, with various needle positions and angles, and achieved a great success rate in every category.

## 2  Related Works

Our work is directly related to deformable linear object (DLO) manipulation, especially the task of robotic insertion. Besides, since our work also relies on the tactile sensor, we also discuss related works on tactile perception-based object manipulation.

**Robotic Insertion of Deformable Linear Object.** Robotic manipulation of deformable linear objects (DLOs) such as cables, wires, threads, or tubes has been a long-standing challenge in the field of robotics. The inherent flexibility of these objects, their interaction with the environment, and the high-dimensional nature of their state space pose unique challenges.

Several manipulation tasks involving DLOs have been explored by previous researchers, such as shape control [9, 10, 11], cable routing [12] and knotting/unknotting [13, 14, 15]. A subset of the research has focused specifically on the task of robotic insertion of DLOs, including needle-threading [1, 2, 3, 16] and DLO-in-hole assembly tasks [17, 18]. Kim et al. [3] adopted a dual camera setting to capture both peripheral and foveated vision, and leveraged imitation learning to train a policy to execute the needle threading task. Since imitation learning requires expert demonstrations, visual servoing is more direct. However, due to the challenge of locating the thread and needle in the common visual setting, many works adopted enhanced visual settings. Silverio et al. [1] adopted a laser scanner to track the tail-end of the thread. However, the laser scanner is not a common choice for general-purpose manipulation settings. Huang et al. [2] employed a high-speed camera to perceive the thread. They assumed that a rapidly rotating thread can be approximated as a rigid object, thereby simplifying the control models. However, this study was based on a specifically designed two-degree-of-freedom mechanism and did not automate the thread mounting process. Lv et al. [16] propose a model-based RL method for the needle-threading task. They adopt differentiable simulation and rendering techniques to synchronize the thread and needle configuration between the simulation and real-world observations. However, they cannot determine whether the execution is successful unless manually examine the execution.

**Tactile perception-based object manipulation.** Tactile perception-based object manipulation is often studied accompanied by the development of tactile sensors. For example, Zanella [18] proposed to address the DLO-in-hole assembly tasks with customized low-resolution tactile sensors on the gripper finger. She et al. [19] explored cable manipulation with a proposed tactile-reactive gripper. Later, as more and more tactile sensors get commercially available or open-source [20, 21, 5], researchers can focus on developing algorithms for different tasks, such as object shape reconstruction [22, 23, 24, 25], contour following [12, 26] and dexterous manipulation [27].

## 3  Background

In this section, we will first describe the task settings and assumptions. Then we describe the problems in the tactile-based needle-threading task.

### 3.1  Task Setting

The overall task setting is illustrated in Fig. 1. In the experiments, we need two robot arms, one for thread manipulation (denoted as *thread robot*), and one for eyelet localization (denoted as *eyelet robot*). To grasp and sense the thread, we mount two tactile sensors on the inner side of the gripper fingers on the *thread robot*. To touch and sense the eyelet, we mount one tactile sensor on the outer side of one gripper finger on the *eyelet robot*. In this work, we adopt Franka Emika panda as the *thread robot* and Kinova Gen2 as the *eyelet robot*. We adopt MC-Tac sensor [4] as our tactile sensors to provide tactile perception. We mount an Intel RealSense D435 on the *thread robot*, and conduct eye-in-hand calibration with Easy-HandEye package [28].

The thread is scrolled on a spool, and a part of the thread is hung down naturally. The spool is controlled by a brake device. When we trace the thread in *Tail-end Finding* stage, the spool will not be able to scroll so that the length of the thread dropped can remain the same. While in *Tail-end Insertion* stage, the spool can rotate so that the thread can reach the eyelet.

The needle is inserted into a base support because we do not consider picking the needle up. The location of the base support and the angle of the needle can be varied.

The tactile sensor on the *eyelet robot* touches the eyelet from the beginning of the experiments, but the exact location of the eyelet is not known.

## 3.2   Problem Statement

In the *Tail-end Finding* stage, given a thread, we first find its rest length $l_{\text{thread}}$ by gliding it from the beginning point to the thread tip. The beginning point is defined at the center of $2cm$ below the spool. Then, we trace the thread from the same beginning point and stop the movement early, so that the tail-end has a length of $l_{\text{tail}}$. To note, when the gripper holds the tail-end and moves around, gravity might cause the tail-end to bend downwards. Thus, we need to estimate the tip position in consideration of the gravity force. We solve this problem with a neural network $f(\cdot)$:

$$p_{\text{tip}} = f(l_{\text{tail}}, \theta, p_{\text{tac}}) + p_{\text{tac}}, \tag{1}$$

where $\theta$ is the orientation of the tail-end measured from the tactile image, $p_{\text{tac}} \in SO(3)$ is the position w.r.t *thread robot*'s base, and $p_{\text{tip}} \in SO(3)$ is the position w.r.t *thread robot*'s base, as shown in Fig. 2.

In the *Tail-end Insertion* stage, we drive the tail towards the eyelet area. We first give a rough estimate of the eyelet area with $p_{\text{eyelet}} = p_{\text{marker}} + T_0$. $p_{\text{marker}}$ is the relative pose from the marker on *eyelet robot* to *thread robot*'s base, which is easy to obtain after eye-in-hand calibration. $T_0$ is the transformation between the marker and the tactile sensor on the *eyelet robot*, which only needs to measure once for all the experiments. Given the estimated $p_{\text{tip}}$ and $p_{\text{eyelet}}$, we plan a trajectory to a location perpendicular to the gel surface and $1cm$ away. In this way, the *thread robot* later need only move along the surface to find the eye and perform insertion, the surface is denoted "$uv$-plane", as shown in Fig. 1.

Then, we read the tactile image stream $I \in \mathbb{R}^{1600 \times 1200 \times 3}$ from the tactile sensor on the *eyelet robot* and model the insertion task as a goal-conditioned reinforcement learning problem. Given an eyelet imprint $I_{\text{eye},t}$, we segment the hole and the thread tip-gel contact point from $I_{\text{eye},t}$ with a pretrained segmentation model, Grounded-SAM [7, 8]. We then calculate the center of mass (COM) of contact point mask $c_{i,t}$, and we randomly sample $N$ pixel positions from the mask of needle eyelet, and obtain $C_{\text{eye},t} = \{n_{\text{eye},t}^1, ..., n_{\text{eye},t}^N\}$. We formulate the problem of *Tail-end Insertion* as learning a policy $\pi$ that sequences move actions $a_t \in \mathcal{A}$ with a robot from tactile observations $(c_{i,t}, C_{\text{eye},t}) \in \mathcal{O}$.

$$\pi(c_{i,t}, C_{\text{eye},t}) \rightarrow a_t = (\mathcal{T}_u, \mathcal{T}_v) \in \mathcal{A} \tag{2}$$

with $\mathcal{T}_u$ and $\mathcal{T}_v$, defined in SE(2), is the displacement of the end-effector of the thread robot in the $u$ $v$ directions respectively. We calculate the total number of pixels of the poking thread, as well as the number that is within the eyelet mask. If the total number is lower than $500$, or the number of steps $t$ that the agent has tried is larger than $5$ (based on the average reward of training), we consider it a **failure**. If the number of pixels that are in the needle surpasses half of the total pixel number, we regard the state as **successful**. One of the steps in the real world is shown in Fig. 3.

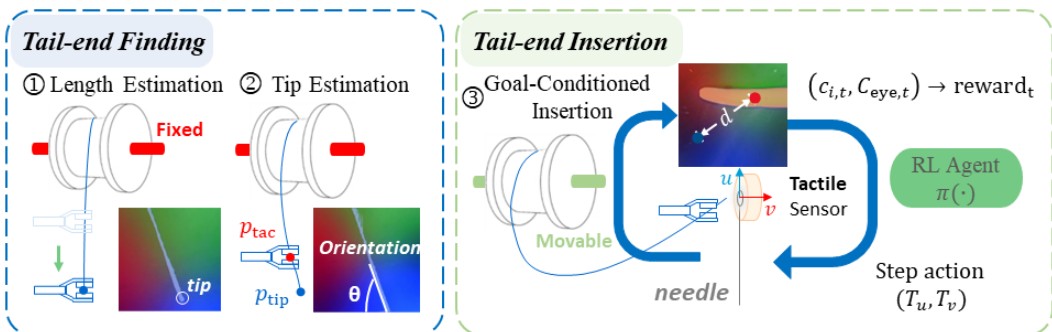

Figure 2: The pipeline of T-NT consists of two aforementioned parts: *Tail-end Finding* and *Tail-end Insertion*. We first do the tip pose estimation with a neural network, and adapt our RL agent trained in a simulation environment to conduct needle threading with the observation from the tactile sensor.

# 4 Method

## 4.1 Tactile Image Processing

For all the tactile images read during the whole process, we adopt the Grounded-SAM [7, 8] model with a prompt of "line" to locate the thread imprint, and "bump" for the poke imprint. As for the eyelet, we use the prompt of "big hole". In our experiments, we find these prompts are robust enough. Thanks to the generalizability of the segmentation model, we don't have to train a specific segmentation model to get the imprint mask for both thread and eyelet.

## 4.2 Tip Pose Estimation

As mentioned in Sec. 3.2, once we trace the thread twice, we can obtain $l_{\text{tail}}$, and then we can estimate the orientation $\theta$ of the tail-end from the tactile image. $l_{\text{tail}} = d1 - d2$ where $d1$ and $d2$ are the distances the gripper traverses in the two times respectively. The orientation $\theta$ of the line is estimated by the incline of the bottom contour, conducted by OpenCV [29], of the line mask in the tactile image, as shown in Fig. 2.

With $l_{\text{tail}}$ and $\theta$, we need to estimate the tip position. According to Equ. 1, we train a 4-layer MLP $f(\cdot)$. To collect the training data, the *thread robot* grasps the tail-end of the thread and moves to a random direction 500 times. Each time, we measure the $l_{\text{tail}}$ and the transformation between $p_{\text{tip}}$ and $p_{\text{tac}}$ manually, while we can automatically obtain the data of $p_{\text{tac}}, \theta$. More details about the neural network structure and its training can be found in the supplementary materials.

## 4.3 Goal-conditioned Insertion

**Training Setup in Simulation**  We adopt a Unity-based simulation environment to train the RL model. We set up the same hardware setting in the simulator and the rendering of tactile images has been implemented and adjusted to resemble the real images (See supplementary materials on our website). We model the needle as a rigid object and attach it to a cube as the base support. The clearances of the eyelets are replicated from the real items. As for the thread, we model it as a constrained particle system with XPBD [6]. It allows for adjustment of the stiffness, thickness, and length of the thread.

We initialize the training by randomly setting the relative positions between the end-effector of the *thread robot* and the eyelet. Considering the calibration error in the real world, the randomized distance is set to at most $3cm$. The *thread robot* moves along the $uv$-plane of the gel surface of the tactile sensor on *eyelet robot*, and tries to poke gel and insert the tip into the eyelet.

As mentioned earlier, to mitigate the sim-to-real gap, instead of utilizing the tactile image as the input to the policy model, we introduce an intermediate representation, *i.e.*, the locations of the eyelet and thread tip calculated by the mask generated on the tactile image from eyelet $I_{\text{eye}}$ with the segmentation model. We model the insertion problem as the goal-conditioned RL based on this intermediate representation of the observation.

The reward function we use is: $\text{reward}_t = \frac{-\sum_{i=1}^{N} d(c_{i,t}, n_{\text{eye},t}^i) \, / \, l}{N}$. When the task is successful, $\text{reward}_t = r$ and when the task is failed, $\text{reward}_t = -r$. With $r = 100$, $l$ is the pixel length of the tactile image diagonal, $d(\cdot)$ is the Euclidean distance. The clear definition of success and failure has been mentioned in Sec. 3.2.

**Transfer to Real World**  After we train our RL model in the simulation environment, we directly apply it in the real world (See Fig. 3). To note, in the real world, to accomplish the full pipeline of insertion, we need to conduct the *Tail-end Finding* process and move the tip around the eyelet in a near-perpendicular direction to the gel surface.

# 5 Experimental Setup

**Tactile Sensors** The tactile sensor we adopt is MC-Tac sensor [4], the gel has a Young's module of 0.123 MPa. It is important when selecting the thread material since the thread needs to be stiffer to leave the imprint on the gel.

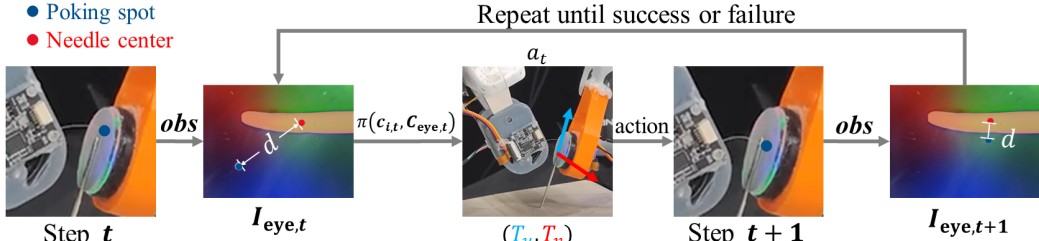

Figure 3: An example of the real-world steps for goal-conditioned insertion. The observations are obtained by the tactile sensor, and the model predicts a $uv$-plane transformation for the thread robot. We repeat the step until the agent judges the state a success or failure.

**Thread-Eyelets** We select three kinds of threads made of nylon, metal, and glass fiber with Young's module of 8.3 GPa, 50 GPa, and 90 GPa respectively.

Aside from the stiffness, the thickness of the threads and the size of the eyelet clearance are also important to show the adaptability of our method. We use threads with thicknesses of $0.2mm$, $0.5mm$, $1mm$, and $2mm$. The sizes of the eyelets clearance are $0.6 \times 7.5mm^2$, $1.6 \times 15mm^2$, and $2.4 \times 9mm^2$. The thread and eyelets we use in the tasks are shown in Fig. 4.

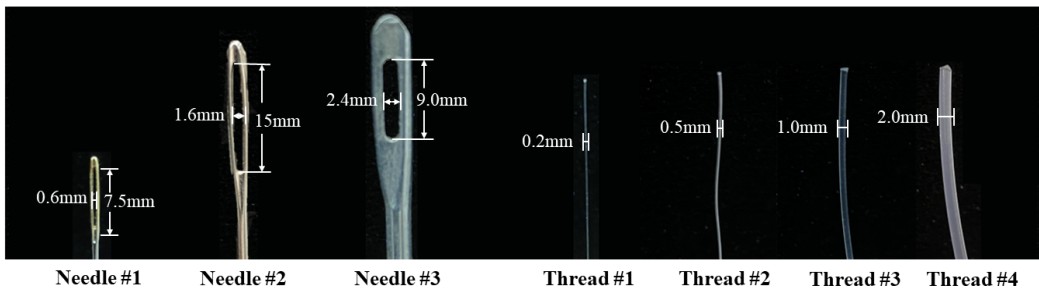

Figure 4: We have selected three kinds of eyelets and four kinds of threads with different sizes, to prove the ability of our method.

As shown in Fig. 5, when we fix the needle onto a base, we vary the angles from $45°$ to $90°$, and the base support positions are randomly placed as long as it is within the reach of the robot.

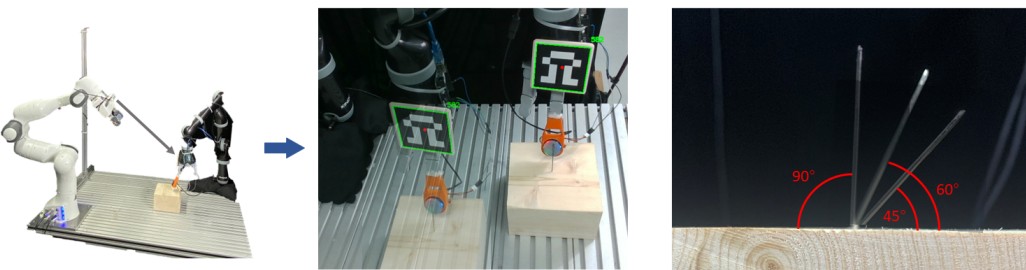

Figure 5: We conduct our experiments on different needle positions and angles, to show the ability of our model to adopt different situations of the needle states.

**Training Details** In our simulation training, we only train our model with needle #2 and line #2. The line in the simulation is modeled with Obi [6], an XPBD-based physics engine. For keeping the line straight, we use the damping value 0.9, and the particle resolution for modeling the line is 0.7. We train the RL model with Stable baselines3 [30] and RFUniverse [31] to interact with Unity. We train three different models: **PPO**, **SAC**, and **DDPG** with the same learning rate of 3e-4 and total timesteps 1e5. The sample number $N$ of needle pixels for reward function calculation is 500.

# 6 Results

## 6.1 Metrics

**Success Rate** If the imprint of the tail-end is inside the eyelet, we consider it a success. We measure the success rate in real-world experiments.

**Mean Distance Error** We measure the distance between the predicted thread tip and the ground truth, and calculate the average distance for every kind of thread.

## 6.2 Evaluation on Tip Pose Estimation

As mentioned earlier, we have prepared 500 data samples for each thread. We split 400 of them to train and 100 to test. We report the quantitative results with mean distance error in Table 1. And the qualitative results are shown in Fig. 6. Since we use the estimated tip position for planning a desired location to insert, thus as long as this location, which is influenced by a sum error of tip position estimation, $p_{eyelet}$ estimation, trajectory following for the robot to execute the plan, is still in the range of the gel surface, the insertion process can continue. And the gel surface has an area of $15mm \times 15mm$, thus the tip pose estimation error is significantly small. Additionally, it takes about **1-2** minutes depending on the length of the thread (5 cm-20 cm).

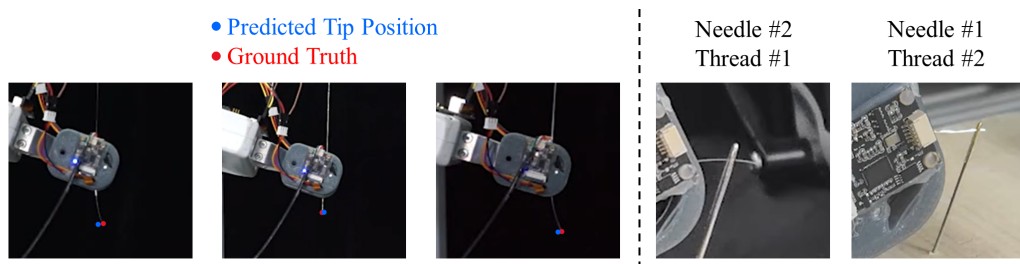

Figure 6: Qualitative results for tail-end estimation and insertion. The tip poses estimation and goal-conditioned insertion allow us to successfully thread needles with different sizes and thicknesses.

| Thread | Mean Distance (mm) |
|---|---|
| **#1** ($\approx 0.2mm$) | 2.2 |
| **#2** ($\approx 0.5mm$) | 1.7 |
| **#3** ($\approx 1mm$) | 3.5 |
| **#4** ($\approx 2mm$) | 4.1 |

Table 1: Quantitative results for tail-end finding.

| Angle | Success Rate |
|---|---|
| 45 | 57.89% |
| 60 | 63.33% |
| 90 | 62.89% |

Table 2: Average success rate for different needle angles.

## 6.3 Results of Tail-end Insertion

We conduct experiments on all three kinds of needle eyelets and four threads, with 20 random base support locations. The needle is fixed on the base support with an angle of $60°$. According to a small test (Table 2) on the angles, it does not have a significant influence on the results.

We have conducted the experiments using a visual servoing method (denoted "VS" in Tab. 3) as a baseline to compare. The method is simply calculating the pixel distance between the COMs of the needle mask and the thread imprint, and projecting it to the real-world distance according to the calibrated camera model, and then planning and moving the gripper accordingly with an off-the-shelf planner, MoveIt! [32].

The quantitative results of the success rate are shown in Tab. 3. Due to needle size, line #3 can't be threaded into needle #1, and line #4 can't be threaded into needle #1 and #2. The results clearly show that, with accurate tail-end finding, the success rates of the *Tail-end Insertion* is significantly high, reaching **63.33%**, while visual servoing method only has **47.22%**, which proves the great performance of our method. Although we only train on one kind of needle and thread, the intermediate representation between real-world tactile image and mask from our simulation environment

successfully solves the sim2real gap and thus provides us great generalizability and transferability for different needles and threads. We can also conclude that the difference in the size of needles and threads affects the success rate greatly, for the bigger needles give the thread more opportunity for trial and error. The difference in the reinforcement learning policy hardly has an impact on the results. The insertion process takes an average of **3.41** steps to complete, with less than **1 minute**. We will elaborate and discuss more factors that influence our experiments and results in the supplementary materials.

| Policy | Thread \ needle | #1 ($\approx 0.2mm$) | #2 ($\approx 0.5mm$) | #3 ($\approx 1mm$) | #4 ($\approx 2mm$) |
|---|---|---|---|---|---|
| VS | **#1** $(0.6 \times 7.5)$ | 35% | 35% | * | * |
| | **#2** $(1.6 \times 15)$ | 55% | 40% | 45% | * |
| | **#3** $(2.4 \times 9)$ | 65% | 50% | 50% | 50% |
| PPO | **#1** $(0.6 \times 7.5)$ | 50% | 45% | * | * |
| | **#2** $(1.6 \times 15)$ | 65% | 65% | 50% | * |
| | **#3** $(2.4 \times 9)$ | 70% | 80% | 85% | 60% |
| SAC | **#1** $(0.6 \times 7.5)$ | 45% | 50% | * | * |
| | **#2** $(1.6 \times 15)$ | 65% | 65% | 45% | * |
| | **#3** $(2.4 \times 9)$ | 80% | 85% | 75% | 65% |
| DDPG | **1#** $(0.6 \times 7.5)$ | 60% | 50% | * | * |
| | **2#** $(1.6 \times 15)$ | 55% | 60% | 55% | * |
| | **3#** $(2.4 \times 9)$ | 75% | 75% | 75% | 60% |

Table 3: Quantitative results for the success rate of four kinds of lines and three different needles, with 3 different RL policy and 20 random base support pose.

## 6.4 Limitation

We briefly analyze the limitation of our method, more analysis and visual failure cases can be found on our website.

**Thread stiffness requirement:** The key limitation of our proposed methodology is its inability to work with soft threads such as the sewing threads. It is due to only the thread which has a stiffness greater than the gel being used can cause the imprint.

**Thread fixation to obtain the length:** Our method takes a two-run process to locate the tail-end of the thread. During the *Tail-end Finding* stage, the thread has to be fixed in one end, ensuring that its remaining length does not change. This could complicate the hardware setting.

**Tactile Sensor Positioning:** The tactile sensor on *eyelet robot* needs to come in contact with the back side of the eyelet. Thus it requires the eyelet to be thin and cannot be applied to applications such as USB insertion where the back side of the hole is not easily accessible. While it can be used for tasks like screw insertion into a nut.

## 7 Conclusion

In this work, we present T-NT, a novel approach to deformable linear object (DLO) insertion tasks, particularly needle threading. Our strategy incorporates a tactile perception-based approach for both *Tail-end Finding* and *Tail-end Insertion* stages, utilizing the vision-based tactile sensor. The reinforcement learning model for the insertion stage, trained in a simulated environment, was shown to be effective and adaptable to real-world scenarios. This tactile-based approach indicates that the use of tactile perception, combined with reinforcement learning, can facilitate research involving highly deformable objects and requires precise manipulation. Further research may benefit from building on this work to extend its application to other DLOs and complex manipulative tasks.

**Acknowledgments**

This work was supported by the National Key R&D Program of China (No. 2021ZD0110704), Shanghai Municipal Science and Technology Major Project (2021SHZDZX0102), Shanghai Qi Zhi Institute, and Shanghai Science and Technology Commission (21511101200).

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
