# OpenReview forum: "Precise Robotic Needle-Threading with Tactile Perception and Reinforcement Learning"
_robot-learning.org/CoRL/2023/Conference — CoRL 2023 Poster_

### Official Review · Reviewer_t9QJ · 2023-07-09

**Confidence:** 3
**Originality:** Good
**Technical Quality:** Good
**Clarity Of Presentation:** Good
**Impact:** 3

**Recommendation:**

Weak Accept: I recommend accepting the paper, but will not argue for my recommendation if the majority of other reviewers have a different opinion.

**Review:**

I thought this was a very difficult robotic task, so I commend the authors for making progress in this setting.

I do not have a background in this problem setting, so my comments are mainly high level.

I was surprised to see there wasn't an appendix? I would have liked to have seen the RL training curves, to see how hard the task was to learn.

I thought the paper was perhaps too acronym heavy. The T-NT acronym is a bit labored and not really needed as there aren't any baselines to differentiate against. The paper introduces 'Grounded-SAM' a few times, and I wonder if it could just be referred to as a segmentation model. It also appears to be language conditioned based on the passage in 4.1 which isn't mentioned in the text previously. The XPBD acronym is not explained (extended position-based dynamics?)

I would have been interested to see a non-RL baseline, perhaps something hand-engineered like a visual servoing feedback law. It would have been good to see what benefit RL added to the performance over human engineering.

OpenCV should be cited in Section 4.2.

It would be good to have a success / failure definition closer to Equation 3.

The paper formatting is a bit mess, with some clear vspace hacking in various places. There is plenty of space optimization that could be done, such as heading 4.1 being unnecessary, and Equation 3 generating a lot of white space. It's also visually cleaner to have figures and tables either on the top or bottom of a page, rather than floating. It also looks like the ```\text``` macro is not being used in any of the math, so the math text appears quite wonky (e.g. $reward_t = \dots $ rather than $\text{reward}_t = \dots $).

**Quality Of The Limitations Section:**

Limitations are addressed clearly

**Questions For Rebuttal:**

None.

**Robotics Focus:**

Sufficient demonstration on hardware

**Summary Of Paper:**

The authors propose a robotic system for threading a needling, using image segmentation and deep reinforcement learning.

**Summary Of Recommendation:**

Impressive systems paper on a complex task. I recommend acceptance.

---

> ### Author Response · Authors · 2023-08-14
> **Revised paper uploaded**
>
> **Update:**
> We have uploaded our revised paper, namely **T_NT_revised.pdf**, in the original **rebuttal.zip** we uploaded through rebuttal.

---

### Official Review · Reviewer_zrs7 · 2023-07-18

**Confidence:** 5
**Originality:** Fair
**Technical Quality:** Fair
**Clarity Of Presentation:** Good
**Impact:** 2

**Recommendation:**

Weak Reject: I recommend rejecting the paper, but will not argue for my recommendation if the majority of other reviewers have a different opinion.

**Review:**

Strengths:

The paper addresses an interesting task -- needle threading. The presentation of the method is relatively clear and straightforward.
There are sufficient quantitative results and real-world experiments.


Weaknesses:

It seems to me the task setup on attaching a tactile sensor to the eyelet robot is a little bit redundant. In reality, whether the thread could pass the needle could be detected from a front-view camera or the tactile sensor in the thread robot part. Also, given that there exist camera and tactile information and estimated states, it is natural to have a visual servo baseline to compare with. Besides, there should be more analysis of the failure cases of not reaching 100% even when the thread is small (0.2) and the needle is large (2.4x9).

Additional point:
Figure2 expresses the algorithm step by step but may need to be approved to make readers understand it easier.


**Quality Of The Limitations Section:**

Limitations are not well addressed

**Questions For Rebuttal:**

As described above:
1) please add more analysis in related works with imitation learning for needle threading (e.g https://www.youtube.com/watch?v=ytpChcFqD5g&ab_channel=UTISILab)
    more analysis in related work with visual servoing methods
2) Why using a second arm with tactile sensing is necessary and important
3) In the second stage, more analysis on why RL method can not reach 100% even when the thread is small and the needle is large
4) Adding imitation learning baseline and visual servoing baseline
5) Show the number of inserting trials for each agent


**Robotics Focus:**

Sufficient demonstration on hardware

**Summary Of Paper:**

The paper presents a learning method for tactile-based needle threading tasks with a dual arm setup. The proposed method T-NT will first locate the thread and then employ a tactile-guided reinforcement learning (RL) model to drive the robot to insert the thread. And the paper shows experiments on real and sim with different sizes of needles and threads.


**Summary Of Recommendation:**

The paper shows a solution for tactile-based needle threading. However, more experiments, details, and clarification needed to be added to show the validity of the method. (as described in previous sections)

Therefore, I am leaning toward rejecting this work but am willing to change my mind if the authors provide a convincing rebuttal.

---

> ### Author Response · Authors · 2023-08-14
> **Revised paper uploaded**
>
> **Update:**
> We have uploaded our revised paper, namely **T_NT_revised.pdf**, in the original **rebuttal.zip** we uploaded through rebuttal.

---

### Official Review · Reviewer_KsbS · 2023-07-19

**Confidence:** 5
**Originality:** Good
**Technical Quality:** Very Good
**Clarity Of Presentation:** Very Good
**Impact:** 4

**Recommendation:**

Strong Accept: I recommend accepting the paper and will argue for my recommendation even if other reviewers hold a different opinion.

**Review:**

Strengths:
- The paper expresses the method clearly and presents the results well.
- Reinforcement learning in simulation environment gives smooth transition to real-world experiements.
- The real-world experiments show that the method is effective and reliable.
- Several different kinds of eyelets and threads are selected to test the adaptability of the method.

Weaknesses:
- Most kinds of threads are soft, and there seems to be no guarantee that it will be detected on the sensors every time. At this point CV seems to provide better recognition.
- There seems to be a gap between the images in the simulation environment and those in the real experiments, and how the impact of this gap is overcome?

**Quality Of The Limitations Section:**

Limitations are addressed clearly

**Questions For Rebuttal:**

1. Most kinds of threads are soft, and there seems to be no guarantee that it will be detected on the sensors every time. At this point CV seems to provide better recognition.
2. There seems to be a gap between the images in the simulation environment and those in the real experiments, and how the impact of this gap is overcome?

**Robotics Focus:**

Sufficient demonstration on hardware

**Summary Of Paper:**

This paper provides a method for tail-end finding and insertion in the needle-threading task. Reinforcement learning in simulated environment gives a smooth transition to real-world applications. The authors also conduct extensive experiments on real robots showing that the proposed method is effective.

**Summary Of Recommendation:**

Great strengths outweighs minor weaknesses. Strong accept.

---

> ### Author Response · Authors · 2023-08-14
> **Revised paper uploaded**
>
> **Update:**
> We have uploaded our revised paper, namely **T_NT_revised.pdf**, in the original **rebuttal.zip** we uploaded through rebuttal.

---

### Official Review · Reviewer_uQnm · 2023-08-02

**Confidence:** 4
**Originality:** Good
**Technical Quality:** Very Good
**Clarity Of Presentation:** Very Good
**Impact:** 3

**Recommendation:**

Weak Accept: I recommend accepting the paper, but will not argue for my recommendation if the majority of other reviewers have a different opinion.

**Review:**

Overall I think the quality of the paper is good. The clarity is also fine. They have cited previous works that used a laser scanner mounted on the end of another Franka robot to inspect where the eyelet is in order to insert it so it is not an original problem, but they used gelsight slims to solve the problem instead of a more expensive laser scanner. They used Grounded-SAM out of the box to locate the features that they cared about in the tactile images which was nice. They used traditional reinforcement learning techniques of PPO, SAC, and DDPG to train with once they had the reward model. I don't see any real weaknesses in the paper. This seems like a paper that uses tactile sensors and many out of the box components connected together in order to accomplish a pretty difficult task.

**Quality Of The Limitations Section:**

Limitations are addressed clearly

**Questions For Rebuttal:**

I don't have any real questions. I just wonder how long one insertion from start to finish would take.

**Robotics Focus:**

Sufficient demonstration on hardware

**Summary Of Paper:**

The authors in this paper developed a pipeline of using a Gelsight slim on both a Franka and a Kinova arm to first trace the end of one thread by gripping and sliding it while it was held in place by an overhead spool with a brake using the Franka and then using the Kinova to press against the eye of a needle to locate it and then classify whether the end of the thread was correctly inserted into the eye. They used Grounded-SAM to determine the location and orientation of the thread and the tip from the tactile images and also to locate the eyelet. Afterwards, they used reinforcement learning in simulation to learn a policy to insert the tip of the thread into the eyelet once they have both been identified. Their end result is around 60% success rate in the real world on various different thread sizes and eyelet sizes.

**Summary Of Recommendation:**

I gave these recommendations because it is more of a systems paper. While I do not think there will be as much impact because no extremely "novel" techniques were introduced, it was a nice demonstration of how to use gelsight sensors to do a task given the state of the art techniques.

---

> ### Author Response · Authors · 2023-08-14
> **Revised paper uploaded**
>
> **Update:**
> We have uploaded our revised paper, namely **T_NT_revised.pdf**, in the original **rebuttal.zip** we uploaded through rebuttal.

---

> ### Comment · Reviewer_uQnm · 2023-08-14
> **Response to Rebuttal**
>
> It takes quite a bit long to find the tail end. I will still maintain my evaluation of weak accept.

---

### Comment · Area_Chair_w1nF · 2023-08-10
**Response to reviewers**

Dear Authors,

please submit your responses to reviewers on Openreview soon to enable a constructive discussion with reviewers during the rebuttal time window.

Best,

AC

---

### Decision · Program_Chairs · 2023-08-30

**Decision:**

Accept (Poster)

**Comment:**

The paper presents a system integration for needle threading demonstrated in simulation and real robot experimentation. The method uses Grounded-SAM to determine the location and orientation of the thread and the tip from the tactile images and also to locate the eyelet. Then, reinforcement learning in simulation learns a policy to insert the tip of the thread into the eyelet once they have both been identified. with a 60% success rate in the real world on different thread sizes and eyelet sizes.

One major concern from the reviews is that the technical contribution is missing since it is mostly a system paper that puts several existing components together. After discussion with reviewers and PCs, we believe that it is a well-written system paper for a very complex task. Its presentation at CoRL is beneficial to the audience and could inspire follow-up works.

Please revise your paper according to the feedback by reviewers, in particular, justify the bi-manual system and (1). why simulating the environment and doing domain randomization is difficult and hard to segment the threads? (2) Why RL agents can not poke the thread's position correctly given the good thread segmentation?